# Unveiling the endogenous CRISPR-Cas system in *Pseudomonas aeruginosa* PAO1

**Javier Alejandro Delgado-Nungaray**[1], **Luis Joel Figueroa-Yáñez**[2], **Eire Reynaga-Delgado**[3], **Ana Montserrat Corona-España**[4], **Orfil Gonzalez-Reynoso**[1] *

1 Chemical Engineering Department, University Center for Exact and Engineering Sciences, University of Guadalajara, Guadalajara, Jalisco, Mexico, 2 Industrial Biotechnology Unit, Center for Research and Assistance in Technology and Design of the State of Jalisco, A.C. (CIATEJ), Guadalajara, Jalisco, Mexico, 3 Pharmacobiology Department, University Center for Exact and Engineering Sciences, University of Guadalajara, Guadalajara, Jalisco, Mexico, 4 Chemical Department, University Center for Exact and Engineering Sciences, University of Guadalajara, Guadalajara, Jalisco, Mexico

* orfil.gonzalez@academicos.udg.mx

## Abstract

Multidrug resistance in *Pseudomonas aeruginosa*, a high-priority pathogen per the World Health Organization, poses a global threat due to carbapenem resistance and limited antibiotic treatments. Using the bioinformatic tools CRISPRCasFinder, CRISPRCasTyper, CRISPRloci, and CRISPRImmunity, we analyzed the genome of *P. aeruginosa* PAO1 and revealed an orphan CRISPR system, suggesting it may be a remnant of a type IV system due to the presence of the DinG protein. This system comprises two CRISPR arrays and noteworthy DinG and Cas3 proteins, supporting recent evidence about the association between type IV and I CRISPR systems. Additionally, we demonstrated a co-evolutionary relationship between the orphan CRISPR system in *P. aeruginosa* PAO1 and the mobile genetic element and prophages identified. One self-targeting spacer was identified, often associated with bacterial evolution and autoimmunity, and no Acr proteins. This research opens avenues for studying how these CRISPR arrays regulate pathogenicity and for developing alternative strategies using its endogenous orphan CRISPR system against carbapenem-resistant *P. aeruginosa* strains.

**Data Availability Statement:** All relevant data are within the manuscript and its Supporting Information files.

## Introduction

Clustered regularly interspaced short palindromic repeats (CRISPR) is an adaptive immune system mechanism present in ∼50% of bacteria that plays a role in regulating virulence factors, which are also regulated by quorum sensing (QS), a cell-cell communication through autoinducers, either in a dependent or independent manner of CRISPR-associated (Cas) proteins [1–3]. The CRISPR-Cas system facilitates interference processes and is commonly used in genetic engineering [4]. This system is classified into two classes, each subdivided into three types, and a total of 33 subtypes [5]. Recently, the endogenous CRISPR-Cas system has emerged as a genome editing tool for bacteria that carry it due to its effectiveness, simplicity, suitability, and cost-efficiency. It also avoids the potential cytotoxicity associated with the use

**Funding:** This research was funded by CONAHCYT and the Doctoral Program in Sciences in Biotechnological Processes at the University of Guadalajara with the scholarship number 1267568. The funders had no role in study design, data collection and analysis, decision to publish, or preparation of the manuscript.

**Competing interests:** The authors have declared that no competing interest exist.

of heterologous Cas effectors and facilitates plasmid transformation because of its smaller size [6, 7].

The World Health Organization [8] has classified *Pseudomonas aeruginosa* as a high-priority pathogen due to its multidrug resistance (MDR) posing a global threat. The most critical form is carbapenem-resistant *P. aeruginosa* (CRPA), particularly in healthcare-associated infections, which has a high mortality rate ($>30\%$) and very limited potential antibiotic treatments.

The *P. aeruginosa* PAO1 strain is widely utilized in genetics research aimed at understanding and combating drug resistance through the study of biofilm formation, QS, and its disruption, known as quorum quenching (QQ) [9]. For these reasons, it is crucial to identify and characterize the endogenous CRISPR arrays and Cas proteins in *P. aeruginosa* PAO1 as an initial step towards understanding its CRISPR-Cas system and developing alternative strategies against CRPA.

A bioinformatic approach is essential for understanding the complex functions and evolution of CRISPR-Cas systems. Genomic pattern-matching programs and machine learning algorithms can predict CRISPR arrays, while classical protein homology search tools like the Hidden Markov Model (HMM) can identify Cas proteins in bacteria. For this reason, a combination of various bioinformatic methods is required to characterize each CRISPR-Cas system and understand its evolutionary immune system [10–12].

The first study of the CRISPR-Cas system in *P. aeruginosa* PAO1, using the bioinformatic tool CRISPRFinder, concluded that this strain does not harbor a CRISPR-Cas system [13]. Moreover, nowadays, CRISPRFinder is an old version that has been updated to CRISPRCasFinder. Wheatley and MacLean [14] utilized this last one, and they computationally identified the presence of I-F, I-E, and I-C as CRISPR-Cas system subtypes in 300 *P. aeruginosa* genomes. However, these studies did not provide precise or additional information concerning punctual CRISPR-Cas details in *P. aeruginosa* PAO1. Uncertain CRISPR structures should not be overlooked, even if they slightly deviate from canonical forms, as many CRISPR structures initially go undetected because of insufficient processing of microbial genomes in databases [15].

CRISPRimmunity, CRISPRloci, CRISPRCasTyper, and CRISPRCasFinder are some of the most updated bioinformatical tools to identify CRISPR-Cas systems from complete genome sequences, where CRISPRimmunity distinguishes from the others due to its robust development by integrating multiple databases and other computational tools (such as PILER-CR, CRT, and CRISPRidentify), also its Cas database is the most up-to-date using an updated version of HMM similar to that utilized by CRISPRCasTyper and CRISPRCasFinder, while CRISPRloci uses the Casboundary method. Each bioinformatics tool relies on specific databases and algorithms, which can introduce biases. Therefore, using multiple tools for a comprehensive analysis is important, as their combined use mitigates individual biases and provides a more thorough overview of CRISPR-Cas systems in *P. aeruginosa* PAO1 [16–19].

Therefore, the use of advanced bioinformatic tools is crucial to elucidate the endogenous CRISPR-Cas system in *P. aeruginosa* PAO1. Additionally, it is important to investigate whether this genome harbors prophage sequences and anti-CRISPR (Acr) proteins capable of inactivating its system. Understanding the fundamental aspects of its CRISPR-Cas system could help to decide the suitability of exogenous CRISPR-Cas for future research aimed at engineering *P. aeruginosa* PAO1, avoiding its endogenous system for biomedical applications towards developing alternative strategies against CRPA, or using its endogenous CRISPR-Cas system as a genome editing tool.

## Materials and methods

### Obtaining the genome of *Pseudomonas aeruginosa* PAO1

For this research, the *Pseudomonas aeruginosa* PAO1 strain was selected, and its genome was obtained from NCBI [20] (https://www.ncbi.nlm.nih.gov/datasets/genome/; accessed on March 20, 2024) with the GenBank accession GCA_000006765.1, name: ASM676v1, which is a complete genome that consists of a circular chromosome (6.3 Mb).

### CRISPR-Cas identification in *P. aeruginosa* PAO1

The identification and comparison of CRISPR arrays and Cas proteins in *P. aeruginosa* PAO1 was obtained by applying the bioinformatical tools: CRISPRCasFinder [16] v4.2.2 (https://crisprcas.i2bc.paris-saclay.fr/), CRISPRCasTyper [17] v1.8.0 (http://cctyper.crispr.dk), CRISPRloci [18] v1.1.0 (http://rna.informatik.uni-freiburg.de), and CRISPRimmunity [19] (http://www.microbiome-bigdata.com/CRISPRimmunity). The default parameters were used in all four bioinformatic tools.

### Validation of CRISPR-Cas system and DR length analysis

In addition to using the four most up-to-date bioinformatic tools mentioned in the previous section, the validation of the CRISPR-Cas system identification was conducted with two well-annotated CRISPR-Cas systems from *P. aeruginosa* strains PA14, which harbors a CRISPR-Cas type I-F, and *P. aeruginosa* PA83, which possesses both types I-F and IV-A, attributed to its chromosome and plasmid, respectively [15, 21, 22]. Following the procedure previously described, their genomes were obtained from NCBI [20] (https://www.ncbi.nlm.nih.gov/datasets/genome/; accessed on September 2, 2024) with GenBank accession numbers GCA_000014625.1 and CP017293.1, respectively.

Moreover, a statistical analysis of Direct Repeats (DR) was performed to determine whether the DR sequences found in *P. aeruginosa* PAO1 align with the length range commonly observed in other organisms. The DR data were obtained from the CRISPRCasdb database [23] (https://crisprcas.i2bc.paris-saclay.fr/). Additionally, DR data from orphan CRISPRs were obtained from the CRISPRimmunity database [19] (http://www.microbiome-bigdata.com/CRISPRimmunity) to compare DR lengths in orphan CRISPR systems in *P. aeruginosa* strains. The analysis was conducted using STATGRAPHICS Centurion 19 software package (version 19.6.03).

### Screening of unknown and neighbor proteins of CRISPR arrays

The neighbor protein sequences of CRISPR arrays and unknown proteins identified with CRISPRimmunity [19] and CRISPRloci [18] were additionally submitted to investigate their conserved domains (functional units in proteins) and refine the protein classification using Batch CD-search [24] (https://www.ncbi.nlm.nih.gov/Structure/bwrpsb/bwrpsb.cgi).

### Analysis of the adaptive immunity of *P. aeruginosa* PAO1

The prediction and characterization of prophage sequences, mobile genetic elements (MGE), self-targeting sequences, and anti-CRISPR (Acr) were used to elucidate the co-evolution of CRISPR in *P. aeruginosa* PAO1 using CRISPRImmunity [19] (http://www.microbiome-bigdata.com/CRISPRimmunity).

## Phylogenetic analysis and the relation between the CRISPR system and prophages

The multiple sequence alignment was performed using MEGA11 [25], incorporating the prophage sequences, CRISPR spacers, and the MGE identified in *P. aeruginosa* PAO1, aligned with CLUSTALW. Additionally, Enterobacteria phage T4 (NC_000866.4) was included as the phage outgroup. The best-fitting DNA model, namely Hasegawa–Kishino–Yano model (HKY)+F+G4, was determined using IQ-TREE 2 [26], which also applied the bootstrap method with 100,000 bootstrap replications to test the phylogeny, enabling faster analysis. The phylogenetic tree was visualized using FigTree [27].

## Results

### Identification of the CRISPR-Cas system in *P. aeruginosa* PAO1

An orphan CRISPR system was identified in *P. aeruginosa* PAO1 using the CRISPRimmunity bioinformatic tool. This system is composed of two CRISPR arrays (one of 114 bp and another of 228 bp length; the first array was also identified using CRISPRCasFinder, with an evidence level 1 indicating a low potential for this CRISPR array) (Table 1).

The noteworthy consensus among CRISPRImmunity, CRISPRloci, and CRISPRCasTyper is the identification of DinG and Cas3 proteins (see S1 Dataset for more details on the identification of the CRISPR-Cas system). Individually, CRISPRimmunity detected six signature genes encoding the accessory proteins DinG, Cas3, RT, WYL, Csa3, DEDDh, and TnsC. Meanwhile, CRISPRloci identified the proteins Cas1, Cas7, Cas8, Cas11, and Csx3 without CRISPR arrays. Moreover, CRISPRCasTyper identified the proteins Csf4, CysH, RecD, and NYN with no CRISPR arrays.

### Validation of CRISPR-Cas system and DR length analysis

The CRISPR-Cas system in *P. aeruginosa* PA14 was identified as a type I-F, consisting of the proteins Cas6, Cas7, Cas5f, Cas8, Cas3-Cas2, Cas1, Csy3, Csy2, and Csy1, using CRISPRImmunity, CRISPRCasFinder, and CRISPRCasTyper, but was not identified with CRISPRloci. For *P. aeruginosa* PA83 the CRISPR-Cas type I-F was identified using all four bioinformatic tools. Additionally, a CRISPR-Cas type IV located on plasmid was identified specifically through CRISPRCasFinder and CRISPRimmunity and includes the proteins Csf5, Csf4, Csf3, Csf2, and Csf1 (additional information on the CRISPR-Cas system of both strains can be found in the S2 Dataset).

**Table 1. CRISPR arrays in *P. aeruginosa* PAO1.**

| | CRISPR array 1 |
|---|---|
| | Location 343146–343259 (bp) |
| CRISPR sequence *<br>DR length: 28 bp | GGCCAAAGAACTGGAGGGCAAGGTATGA**GCCAGCACAGCGTATCCCGCTCCGTATCCCGTTCAACAGGCTCTCCATCCAGAAGCGT**GGCCAAAGAACTGGAGGGCAAGGTATGA |
| | CRISPR array 2 |
| | Location 3841728–3841955 (bp) |
| CRISPR sequence *<br>DR length: 50 bp | CCCGGCGTAGGACGAATAACCGC*CAGCTCGGCAGAGGCTACGGCGGATAACGCCTGGGGGCGTTATTCGCCCTACCG*CTTGCTCCCTCCCCGTGTAGGGCGAATAACCGCTACGCGGTTATCCGCCA***CCTCGGCAGGCGTTTCGGCGGATAACGCCTGGGGCGTTATTCGCCCTACCG**CTTGCTCCCTCCCCGTGTAGGGCGAATAACCGCTACGCGGTTATCCGCCA |

* **Spacer-1**, DR Consensus, *Spacer-2*, and **Spacer-3** (respectively).

Regarding the statistical analysis of DR length, the analysis of 28,712 DR sequences in several organisms yielded an average DR length of 31.57 bp, with a standard deviation of 4.23 bp, a median of 30 bp, and a minimum and maximum length of 23 bp and 50 bp, respectively. Among these, 4,225 DR sequences had a length of 28 bp, matching the DR length in CRISPR array 1 of *P. aeruginosa* PAO1, and four DR sequences had a length of 50 bp, corresponding to the DR length in CRISPR array 2. In contrast, the analysis of 122 *P. aeruginosa* strains identified with orphan CRISPR systems, encompassing 1,809 DR sequences, resulted in an average DR length of 29.54 bp, with a standard deviation of 6.85 bp, and a minimum and maximum length of 15 bp and 55 bp, respectively. Of these, 1,201 DR sequences in *P. aeruginosa* strains had a length of 28 bp, consistent with the DR in CRISPR array 1 of *P. aeruginosa* PAO1, and 56 DR sequences with the length of 50 bp, corresponding to the DR length in CRISPR array 2 (see S3 Dataset for more details on DR length).

## Screening of unknown and neighbor proteins of CRISPR arrays

In the genome of *P. aeruginosa* PAO1, CRISPRloci detected four unknown proteins as possible candidates for Cas proteins (S4 Dataset). These sequences identified have the conserved domains of RimL, MiaB, HupE/UreJ, FecR, and DUF4880. On the other hand, CRISPRImmunity threw 40 sequences as possible Cas proteins, 20 for each CRISPR array identified, of which only 35 have conserved domains: PvdQ, potG superfamily, 2-thiour_desulf, AcrR, agmatine_a-guB, Arabinose_bd, AraC, COG3224, COG3450, COG3492, DadA, DUF6160 superfamily, DUF6436, EGL9, folX, GlnA, HisM, HopJ, HTH_18, hydroxyacyl-CoA-like_DH_SDR_c-like, LrgB, MenH, MopI, PBP2_PotF, Peptidase_C26, PldB, PRK05723, PRK06483, PRK07044, PRK07480, PRK08259, PRK12606, ssuB, transpos_IS110 superfamily and YohJ.

## Analysis of the adaptive immunity of *P. aeruginosa* PAO1

Bioinformatic analysis of *P. aeruginosa* PAO1 reveals the presence of one MGE, one self-targeting spacer, four prophages (Table 2), and no Acr proteins. The MGE found in *P. aeruginosa* PAO1 corresponds to the *Pseudomonas* phage vB_Pae_CF23b (MK511017.1), with the protospacer located at 32–85 (bp) that matches with the Spacer-2 (92.6% identity). According to this result, Spacer-2 is associated with the adaptation of the CRISPR-Cas system in *P. aeruginosa* PAO1, where the MGE from *Pseudomonas* phage vB_Pae_CF23b was incorporated into CRISPR array 2. The match between protospacer from the *Pseudomonas* phage vB_Pae_CF23b (MK511017.1) and Spacer-2 is shown below:

CAGCTCGGCAGAGGCTACGGCGGATAACGCCTGGGGGCGTTATTCGCCCTACCG Spacer-2
CCCCACAGCAGAGGCTACGGCGGATAACGCCTGGGGGCGTTATTCGCCCTACCG
Protospacer

*P. aeruginosa* PAO1 harbors one self-targeting spacer in which the protospacer matches with an identity of 100% with the CRISPR Spacer-3 identified in previous subtitles. The protospacer is located at 3843530–3843580 (bp), with a length of 51 bp. Given that the self-targeting spacer does not resemble the prophage sequences detected, *P. aeruginosa* PAO1 did not

**Table 2. Prophage identification in *P. aeruginosa* PAO1.**

| Sequence | Location (bp) | Closest phage | E-value | Identity (%) |
|---|---|---|---|---|
| 1 | 673190–705966 | Phage PS17 (D26449.1) | $2.0e^{-188}$ | 83.2 |
| 2 | 789143–796776 | *Pseudomonas* phage Pf1 (NC_001331.1) | $7.9e^{-59}$ | 100.0 |
| 3 | 2947802–2954696 | uncultured *Caudovirales* phage (NC_055725.1) | $1.5e^{-90}$ | 82.1 |
| 4 | 4051563–4060625 | *Pseudomonas* phage DVM-2008 (EU982300.1) | $8.1e^{-57}$ | 76.0 |

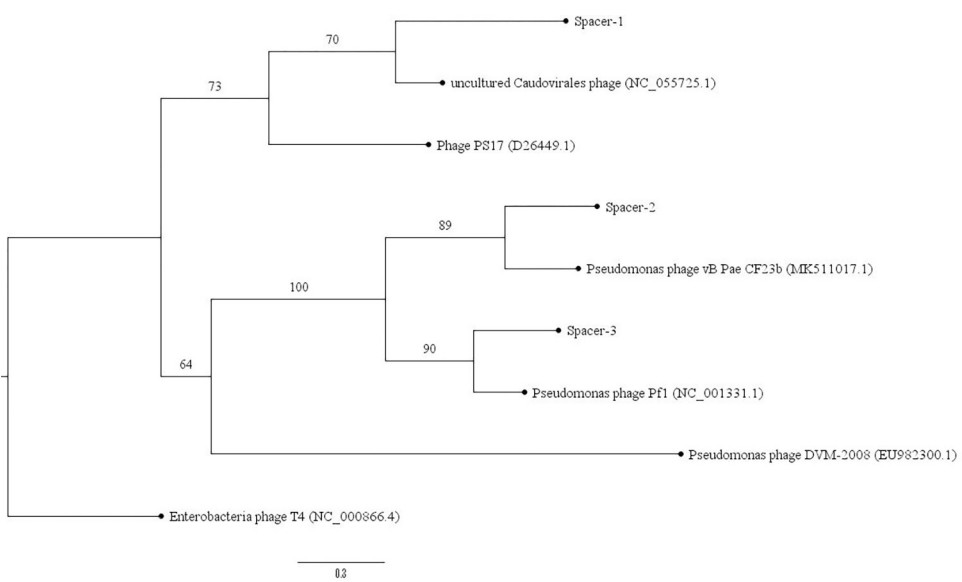

**Fig 1. Maximum likelihood bootstrap phylogenetic tree of CRISPR spacers, prophages, and MGE in *P. aeruginosa* PAO1.** Enterobacteria phage T4 was included to serve as the phage outgroup for phylogenetic rooting. Constructed with Hasegawa–Kishino–Yano model (HKY)+F+G4 and 100,000 bootstrap replicates. Evolutionary analyses were conducted in IQ-TREE 2.

incorporate it into its genome from those prophages. The sequence of the self-targeting spacer found is CCTCGGCAGGCGTTTCGGCGGATAACGCCTGGGGCGTTATTCGCCCT ACCG.

## Phylogenetic analysis of relation between the CRISPR spacers, prophages, and MGE

The phylogenetic tree was constructed to infer whether the identified prophages share a common origin and homology. The genome sequences of the four prophages Phage PS17 (D26449.1), *Pseudomonas* phage Pf1 (NC_001331.1), an uncultured *Caudovirales* phage (NC_055725.1), and *Pseudomonas* phage DVM-2008 (EU982300.1), along with the CRISPR-Cas spacers found in *P. aeruginosa* PAO1, were used to construct the phylogenetic tree, as shown in Fig 1.

Spacer-2 and the MGE identified, which corresponds to the *Pseudomonas* phage vB_Pae_CF23b (MK511017.1), show a high resolution between these sequences (89%). Moreover, the evolutionary history of Spacer-3, which matches the protospacer of the self-targeting spacer, is clearly evolutionary-associated (90%) with *Pseudomonas* phage Pf1 (NC_001331.1). Additionally, Spacer-1 has similarities (70%) with uncultured *Caudovirales* phage (NC_055725.1); it is also revealing that the prophage NC_001331.1 and the MGE MK511017.1 belong to similar evolutionary origins. On the other hand, Phage PS17 (D26449.1), uncultured *Caudovirales* phage (NC_055725.1), and *Pseudomonas* phage DVM-2008 (EU982300.1) belong to distinct lineages within the family of bacteriophages.

## Discussion

The orphan CRISPR system identified in *P. aeruginosa* PAO1 is suggested to be maintained to perform relevant biological functions, such as the regulation of other gene expression, although these functions are not yet fully understood [28]. The DRs identified in the two

CRISPR arrays are within the range of 23 to 50 bp, as observed in the analysis of 28,712 DR sequences. Additionally, these DRs are more aligned with the DR length in orphan CRISPR systems in *P. aeruginosa* strains, which have a wider length range of up to 55 bp, consistent with literature reports that indicate a maximum DR length of 55 bp for bacteria [29, 30]. The identification of the CRISPR-Cas system using the four bioinformatic tools was validated with *P. aeruginosa* PA14 and PA83. The results for these strains are consistent with the literature, showing a CRISPR-Cas type I-F in *P. aeruginosa* PA14 and both a type I-F on the chromosome and a type IV-A on the plasmid in *P. aeruginosa* PA83) [15, 21, 22].

It is possible that this orphan CRISPR system in *P. aeruginosa* PAO1 employs the proteins surrounding the CRISPR arrays as the Cas machinery [31, 32]. Remarkably, the detected Cas proteins include DinG and Cas3, supporting recent evidence about the association between type IV and I CRISPR systems [33]. DinG is deeply associated with CRISPR-Cas IV-A by mediating RNA-guided interference against plasmids; on the other hand, Cas3 is categorized as CRISPR-Cas type I and is generally bound to the CRISPR-associated complex for antiviral defense (Cascade) [14, 15, 22, 34, 35].

The other Cas proteins identified are RT, WYL, Csa3, and DEDDh, which are CRISPR ancillary effectors, and TnsC, which is an effector transposon [5, 19, 36]. Cas1 is required for CRISPR-Cas adaptive immunity and is one of the most evolutionarily conserved components of all CRISPR-Cas systems [37]. Cas7, involved in CRISPR-Cas type IV, and Cas11 are implicated in locking the target DNA after it hybridizes with the CRISPR RNA (crRNA) [38]. Cas8 is involved in immune activation and regulation [39]. Csx3 acts as a ring nuclease identified in type IV and III CRISPR-Cas systems [40, 41]. Csf4, which pertains to the DinG family, and RecD are helicases; CysH possibly acts as a putative effector, all pertaining to CRISPR-Cas type IV systems [33, 42, 43]. The nuclease NYN has implications for crRNA maturation; nonetheless, it is important to establish that this protein has not been well explored [44, 45].

Despite the identification of Cas1, which is involved in spacer acquisition, this CRISPR system lacks a complete Cas1-Cas2 adaption module. Additionally, a co-evolutionary relationship has been demonstrated between the MGE and prophages identified and the orphan CRISPR system in *P. aeruginosa* PAO1. This is evidenced by the detection of protospacers integrated as spacers in the CRISPR arrays, indicating a dynamic role in defense against bacteriophages; both are distinctive features of type IV CRISPR-Cas systems [46]. These new findings contrast with previous studies on the CRISPR-Cas system of *P. aeruginosa* PAO1, which utilized only one computational tool and had fewer databases for comparing CRISPR arrays and Cas proteins [13, 14].

In accordance with the conserved domains identified, PvdQ is a Quorum Quenching enzyme that degrades the autoinducers of QS, suggesting a potential regulation of QS by the CRISPR system through the activation of PvdQ [47]. In the case of the potG superfamily, potG has been declared an off-target integration site that can increase the crRNA expression levels and strengthen the immune response. HopJ, transpos_IS110 superfamily, AraC, and DadA act in some way on the transcription and packaging processes of genetic material, which may be closely related to the adaptive mechanisms of the microorganism for the development of resistance to antibiotics and virulence factors [48–52].

Furthermore, RimL acetylates ribosomal proteins at the $N^{\alpha}$ terminal and is involved in CRISPR-related activities, though its precise function remains unclear [53, 54]. Based on a CRISPR-based dataset, it has been suggested that the tRNA modification enzyme MiaB plays a significant role in cellular functions [55]. It is important to highlight that the purposes of some of the identified domains remain unknown, necessitating further studies.

Notably, the Spacer-2 is associated with the adaptation of the CRISPR-Cas system in *P. aeruginosa* PAO1, where the *Pseudomonas* phage vB_Pae_CF23b, with no Acr protein detected,

was incorporated into CRISPR array 2, indicating that its integration into the CRISPR-Cas system confers immunity against this specific phage without the capability to inactivate it [56–58]. This spacer could be integrated through naïve acquisition from RNA due to the presence of the adaptor Cas1. This mechanism aligns with the acquisition found in type I-E and IV-A CRISPR-Cas systems. Self-targeting spacers are often associated with bacterial evolution, auto-immunity, genome remodeling, and CRISPR-Cas inactivation [59, 60].

These findings reinforce the accurate theory that CRISPR-Cas systems act as a form of acquired immunity and suggest potential gene exchange events between CRISPR spacers in *P. aeruginosa* PAO1 and the prophages and the MGE found in its genome [11, 61]. The self-targeting spacer identified in *P. aeruginosa* PAO1 opens the potential for further investigations into its use for phage therapy by using the closely related phage (*Pseudomonas* phage Pf1) of the self-targeting spacer (Spacer-3) to activate the CRISPR system, thereby inhibiting growth and exacerbating lysis through the combined action of the phage and the toxicity of the self-targeting spacer [62–64].

This orphan CRISPR system appears to be more closely related to CRISPR-Cas class 1 type IV, associated with type I due to the presence of Cas3, and potentially evolved from type III due to the presence of Csx [46]. This system could be a remnant of a decaying CRISPR-Cas system or that Cas protein loss was caused by interaction between the bacterium and a foreign MGE [5]. This theory is supported by the recent categorization of new *P. aeruginosa* strains, such as *P. aeruginosa* PA83, as CRISPR-Cas type IV and the emerging findings about the co-option of type IV and I CRISPR-Cas systems between hosts and MGEs to overcome limitations [22, 65]. These results indicate a complex regulatory network for the CRISPR system in *P. aeruginosa* PAO1 without sufficient available information to consider it an effective CRISPR-Cas system. However, literature highlights the presence of functional CRISPR systems, even when they are incomplete [66].

## Conclusions

This research identified that *P. aeruginosa* PAO1 possesses an orphan CRISPR system, comprising two CRISPR arrays with noteworthy DinG and Cas3 proteins, one self-targeting spacer, and no Acr. The orphan CRISPR system found suggest it may be remnants of a decaying type IV CRISPR-Cas system, with the loss of Cas2 in the adaptive module potentially due to interaction between the bacterium and foreign MGEs. There is a possibility that the neighboring proteins of the CRISPR arrays could function as CRISPR effector proteins. These findings suggest that the endogenous CRISPR system of *P. aeruginosa* PAO1 will not interfere with the use of an exogenous class 2 CRISPR-Cas system for genetic engineering. Furthermore, this opens the possibility for future research into how the CRISPR arrays in *P. aeruginosa* PAO1 regulate its pathogenicity, the precise evolutionary context of its orphan CRISPR system, and further genome editing tools using its endogenous orphan CRISPR system to explore alternative strategies against carbapenem-resistant *P. aeruginosa* strains.

## Supporting information

**S1 Dataset. Identification of CRISPR-Cas system through CRISPRCasFinder, CRISPRCas-Typer, CRISPRloci, and CRISPRimmunity.**
(XLSX)

**S2 Dataset. Validation of CRISPR-Cas identification with *P. aeruginosa* PA14 and PA83.**
(XLSX)

**S3 Dataset. DR length in CRISPR-Cas and orphan CRISPR systems in *P. aeruginosa* strains.**
(XLSX)

**S4 Dataset. Cas and neighbor proteins of CRISPR arrays through CRISPRloci and CRIS-PRimmunity.**
(XLSX)

**S1 Graphical abstract.**
(TIF)

## Author Contributions

**Conceptualization:** Javier Alejandro Delgado-Nungaray, Luis Joel Figueroa-Yáñez, Orfil Gonzalez-Reynoso.

**Data curation:** Ana Montserrat Corona-España.

**Formal analysis:** Javier Alejandro Delgado-Nungaray, Luis Joel Figueroa-Yáñez, Eire Reynaga-Delgado, Ana Montserrat Corona-España, Orfil Gonzalez-Reynoso.

**Investigation:** Javier Alejandro Delgado-Nungaray.

**Methodology:** Javier Alejandro Delgado-Nungaray, Eire Reynaga-Delgado.

**Resources:** Orfil Gonzalez-Reynoso.

**Supervision:** Luis Joel Figueroa-Yáñez, Orfil Gonzalez-Reynoso.

**Writing – original draft:** Javier Alejandro Delgado-Nungaray.

**Writing – review & editing:** Javier Alejandro Delgado-Nungaray, Luis Joel Figueroa-Yáñez, Eire Reynaga-Delgado, Ana Montserrat Corona-España, Orfil Gonzalez-Reynoso.

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
