## [Decision Letter · Decision Letter 0]

30 Aug 2024

PONE-D-24-28894Unveiling the Endogenous CRISPR-Cas System in * Pseudomonas aeruginosa*  PAO1.PLOS ONE

Dear Dr. Gonzalez-Reynoso,

Thank you for submitting your manuscript to PLOS ONE. After careful consideration, we feel that it has merit but does not fully meet PLOS ONE’s publication criteria as it currently stands. Therefore, we invite you to submit a revised version of the manuscript that addresses the points raised during the review process.

Please address both reviewers' comments thoroughly. The experimental validation comment may be ignored but need some brief discussion on why not needed.

We look forward to receiving your revised manuscript.

Kind regards,

Yanbin Yin

Academic Editor

PLOS ONE

Journal Requirements:

1. When submitting your revision, we need you to address these additional requirements. Please ensure that your manuscript meets PLOS ONE's style requirements, including those for file naming. The PLOS ONE style templates can be found at https://journals.plos.org/plosone/s/file?id=wjVg/PLOSOne_formatting_sample_main_body.pdf and https://journals.plos.org/plosone/s/file?id=ba62/PLOSOne_formatting_sample_title_authors_affiliations.pdf 2. Thank you for stating the following financial disclosure:  "Fellowship from Conahcyt with number 1267568"Please state what role the funders took in the study.  If the funders had no role, please state: "The funders had no role in study design, data collection and analysis, decision to publish, or preparation of the manuscript."If this statement is not correct you must amend it as needed. Please include this amended Role of Funder statement in your cover letter; we will change the online submission form on your behalf.

Reviewers' comments:

Reviewer's Responses to Questions

**Comments to the Author**

1. Is the manuscript technically sound, and do the data support the conclusions?

Reviewer #1: Partly

Reviewer #2: Partly

2. Has the statistical analysis been performed appropriately and rigorously? 

Reviewer #1: No

Reviewer #2: No

3. Have the authors made all data underlying the findings in their manuscript fully available?

Reviewer #1: Yes

Reviewer #2: Yes

4. Is the manuscript presented in an intelligible fashion and written in standard English?

Reviewer #1: Yes

Reviewer #2: Yes

5. Review Comments to the Author

Reviewer #1: The mechanism of bacterial adaptive immunity remains incompletely understood. Investigating CRISPR-Cas systems in bacterial genomes and their connections with prophages, MGEs, and anti-CRISPR elements offers a novel approach to understanding CRISPR co-evolution in bacteria. This study identifies an orphan CRISPR system in the genome of Pseudomonas aeruginosa PAO1, suggesting it may function endogenously and regulate pathogenicity.

While the research is suitable for publication, the reviewer has several comments on the manuscript:

1. Given the rapid evolution of microorganisms, the genomic arrangements, protein sequences, and CRISPR arrays of CRISPR-Cas systems can vary significantly among individual strains. Analyzing only one assembly of a Pseudomonas aeruginosa strain is insufficient to draw a robust conclusion. More assemblies should be included for statistical validity.

2. The annotation of CRISPR repeats is not sufficiently precise. Most known CRISPR systems have repeat lengths around 36 nucleotides (nt), but the author has annotated two CRISPR arrays with direct repeat (DR) lengths of 28 bp and 50 bp. These unusual repeat lengths require further explanation.

3. Since genomic arrangement is crucial to the characterization of CRISPR-Cas systems, the reviewer recommends providing the genomic arrangements of several orphan CRISPR systems in Pseudomonas aeruginosa strains to better elucidate the generative characteristics of these systems.

4. The author proposes several hypotheses regarding the function and immune strategies of the newly identified orphan CRISPR-Cas system. However, these inferences lack concrete experimental evidence, which is needed to support the claims.

Reviewer #2: 1. Multiple up-to-date computational tools were chosen for identifying CRISPR-Cas systems. How do these tools compare with others available in terms of sensitivity, specificity, and accuracy on this specific strain? The parameter settings of the computational tools were not justified. Did the sensitivity analyses be performed to determine how changes in parameters affect the outcomes? Are there any potential biases in the computational tools used?

2. For the phylogenetic tree in Figure.1, it may include relevant outgroups or reference prophages for more meaningful evolutionary relationships.

3. A broader comparative analysis might be included by comparing the CRISPR-Cas systems in different P. aeruginosa strains.

4. While this study provides a computational analysis of CRISPR-Cas systems, it would be beneficial to consider future experimental validation (e.g., PCR) to confirm the predictions, which would enhance the credibility of the findings.

6. PLOS authors have the option to publish the peer review history of their article (what does this mean?). If published, this will include your full peer review and any attached files.

Reviewer #1: No

Reviewer #2: No

---

## [Author Response · Author response to Decision Letter 0]

23 Sep 2024

Manuscript ID: PONE-D-24-28894

Title: Unveiling the Endogenous CRISPR-Cas System in Pseudomonas aeruginosa PAO1

Authors: Javier Alejandro Delgado-Nungaray, Luis Joel Figueroa-Yáñez, Eire Reynaga-Delgado, Ana Montserrat Corona-España, and Orfil Gonzalez-Reynoso

Dear Academic Editor Dr. Yanbin Yin and Reviewers,

We would like to express our gratitude for your professional and thorough peer review of our manuscript. We have revised our manuscript and considered all of your suggestions, which have significantly contributed to refining our work. We believe that, thanks to your valuable comments, the revised version now meets the high standards of the PLOS ONE.

Sincerely,

The authors

Reviewers’ Comments and Authors’ Responses

In the italic text: the reviewer’s comments.

In bold notation: authors’ responses

In red notation: major changes incorporated in the manuscript. 

Reviewer #1: The mechanism of bacterial adaptive immunity remains incompletely understood. Investigating CRISPR-Cas systems in bacterial genomes and their connections with prophages, MGEs, and anti-CRISPR elements offers a novel approach to understanding CRISPR co-evolution in bacteria. This study identifies an orphan CRISPR system in the genome of Pseudomonas aeruginosa PAO1, suggesting it may function endogenously and regulate pathogenicity. While the research is suitable for publication, the reviewer has several comments on the manuscript:

1. Given the rapid evolution of microorganisms, the genomic arrangements, protein sequences, and CRISPR arrays of CRISPR-Cas systems can vary significantly among individual strains. Analyzing only one assembly of a Pseudomonas aeruginosa strain is insufficient to draw a robust conclusion. More assemblies should be included for statistical validity.

We appreciate the reviewer’s understanding of the relevance of our investigation into CRISPR co-evolution in bacteria as an approach to study new ways to regulate the pathogenicity of P. aeruginosa PAO1. We also thank for pointing out the statistical validation, as our study focused solely on the strain P. aeruginosa PAO1. However, as we redacted in the introduction of our manuscript, the analysis of multiple assembled genomes often has overlooked CRISPR structures, particularly in this strain, due to a combination of factors including the rapid increase in knowledge about CRISPR-Cas systems and the absence of a comprehensive bioinformatic tool that integrates the most validated methods for CRISPR identification. To address this limitation, we utilized multiple updated and validated bioinformatic tools in a careful and thorough manner to investigate the CRISPR system in P. aeruginosa PAO1.

Nevertheless, we value your suggestion and have included two additional strains, P. aeruginosa PA14 and PA83, to enhance the robustness of our study. This information was incorporated on the subsection Validation of CRISPR-Cas system and DR length analysis (page 4, lines 95-103), on Results section (page 7, lines 146-154), and Discussion section (page 10, lines 214-228). A new dataset (namely S2 Dataset) was included with the data of these assemblies of P. aeruginosa obtained.

In addition to using the four most up-to-date bioinformatic tools mentioned in the previous section, the validation of the CRISPR-Cas system identification was conducted with two well-annotated CRISPR-Cas systems from P. aeruginosa strains PA14, which harbors a CRISPR-Cas type I-F, and P. aeruginosa PA83, which possesses both types I-F and IV-A, attributed to its chromosome and plasmid, respectively [15,21,22]. Following the procedure previously described, their genomes were obtained from NCBI [20] (https://www.ncbi.nlm.nih.gov/datasets/genome/; accessed on September 2, 2024) with GenBank accession numbers GCA_000014625.1 and CP017293.1, respectively.

2. The annotation of CRISPR repeats is not sufficiently precise. Most known CRISPR systems have repeat lengths around 36 nucleotides (nt), but the author has annotated two CRISPR arrays with direct repeat (DR) lengths of 28 bp and 50 bp. These unusual repeat lengths require further explanation.

We appreciate your comment, as noted, the DR lengths of 28 bp and 50 bp were identified in our analysis, and these lengths are within the documented range of 20 to 50 bp, with the maximum DR length in bacteria reported of 55 bp and an average length of 31.32 bp (1–3) (these references are provided at the end of this authors’ response). However, we recognize the opportunity to update our understanding of the DR length range and average with more recent data and to determine whether the DR sequences found in P. aeruginosa PAO1 align with the length range commonly observed in other organisms. Therefore, we performed a statistical analysis of DR lengths, which is included on the subsection Validation of CRISPR-Cas system and DR length analysis (please see page 4, lines 104-111), as well as on the results (page 7, lines 155-166) and discussion (page 10, lines 220-224). A new dataset (namely S3 Dataset) was included with the data of these DR length.

Moreover, a statistical analysis of Direct Repeats (DR) was performed to determine whether the DR sequences found in P. aeruginosa PAO1 align with the length range commonly observed in other organisms. The DR data were obtained from the CRISPRCasdb database [23] (https://crisprcas.i2bc.paris-saclay.fr/). Additionally, DR data from orphan CRISPRs were obtained from the CRISPRimmunity database [19] (http://www.microbiome-bigdata.com/CRISPRimmunity) to compare DR lengths in orphan CRISPR systems in P. aeruginosa strains. The analysis was conducted using STATGRAPHICS Centurion 19 software package (version 19.6.03).

References:

1. O. S. Alkhnbashi et al., Nucleic Acids Res. 49, W125–W130 (2021).

2. M. Buyukyoruk, W. S. Henriques, B. Wiedenheft, CRISPR Journal. 6, 216–221 (2023).

3. R. Ge et al., Sci Rep. 6 (2016), doi:10.1038/srep32942.

3. Since genomic arrangement is crucial to the characterization of CRISPR-Cas systems, the reviewer recommends providing the genomic arrangements of several orphan CRISPR systems in Pseudomonas aeruginosa strains to better elucidate the generative characteristics of these systems.

Regarding your comment, we added the archive of supplemental data “S3 Dataset. DR length in CRISPR-Cas and orphan CRISPR systems in P. aeruginosa strains” that includes the arrangements of all orphan CRISPR systems in this species. This data was obtained as part of the subsection Validation of CRISPR-Cas system and DR length analysis (please see page 4, lines 107-111), as well as on the results (page 7, lines 160-166) and discussed in page 10, lines 220-224).

4. The author proposes several hypotheses regarding the function and immune strategies of the newly identified orphan CRISPR-Cas system. However, these inferences lack concrete experimental evidence, which is needed to support the claims.

We appreciate and agree with your comment. However, we would like to emphasize that the aim of this article was to unveil the CRISPR system in P. aeruginosa PAO1 through bioinformatic analysis. The discussion, as noted, was based on formulating hypotheses that were carefully constructed from the current literature and the significant findings of our research. These hypotheses are intended as a starting point for understanding the potential functions of the orphan CRISPR-Cas system. As highlighted in the manuscript, we state the need for further research to investigate the proposed functions of the orphan CRISPR system in P. aeruginosa PAO1 (please see page 1, lines 26-28 and page 12, lines 302-306) where we outline future directions for experimental research.

Reviewer #2: 

1. Multiple up-to-date computational tools were chosen for identifying CRISPR-Cas systems. How do these tools compare with others available in terms of sensitivity, specificity, and accuracy on this specific strain? The parameter settings of the computational tools were not justified. Did the sensitivity analyses be performed to determine how changes in parameters affect the outcomes? Are there any potential biases in the computational tools used?

We appreciate your insightful feedback regarding the selection of computational tools, which were chosen based on their up-to-date CRISPR arrays, Cas proteins databases. Their validation has been published in some of the most renown journals, and they distinguished by their own methods as described on page 3, lines 64-70. Regarding sensitivity, specificity, and accuracy, each bioinformatic tool reported its own results in its respective scientific paper (references on page 16, lines 373-386, numbered as references 16 to 19). In summary, they are capable of identifying CRISPR-Cas systems, by using their own method, for both bacterial and archaeal organisms. The default parameters for these four bioinformatics tools were used to identify validated CRISPR-Cas systems across a broad range of bacterial and archaeal organisms. This information was added on page 4, lines 89-90:

The default parameters were used in all four bioinformatic tools.

Regarding the biases of bioinformatics tools, our text addresses this on page 2, lines 52-54. To further clarify this, we have added the following statement (please see page 3, lines 70-73)

Each bioinformatics tool relies on specific databases and algorithms, which can introduce biases. Therefore, using multiple tools for a comprehensive analysis is important, as their combined use mitigates individual biases and provides a more thorough overview of CRISPR-Cas systems in P. aeruginosa PAO1.

2. For the phylogenetic tree in Figure.1, it may include relevant outgroups or reference prophages for more meaningful evolutionary relationships.

We agree and appreciate your valuable suggestion. In light of this, we have included the Enterobacteria phage T4 as the phage outgroup. This has been included in the Materials and Methods section and modified according to computational requirements we needed and in Results according with the new results obtained (please see page 5, line 126-130 and Figure 1).

Additionally, Enterobacteria phage T4 (NC_000866.4) was included as the phage outgroup. The best-fitting DNA model, namely Hasegawa–Kishino–Yano model (HKY)+F+G4, was determined using IQ-TREE 2 [26], which also applied the bootstrap method with 100,000 bootstrap replications to test the phylogeny, enabling faster analysis. The phylogenetic tree was visualized using FigTree [27].

3. A broader comparative analysis might be included by comparing the CRISPR-Cas systems in different P. aeruginosa strains.

We thank for suggesting a comparative analysis; however, our study focused solely on the strain P. aeruginosa PAO1. As stated in the introduction of our manuscript, the analysis of multiple assembled genomes often overlooked CRISPR structures, particularly in this strain, due to a combination of factors, including the rapid increase in knowledge about CRISPR-Cas systems and the lack of a comprehensive bioinformatic tool that integrates the most validated methods for CRISPR identification. To address this limitation, we utilized multiple updated and validated bioinformatic tools in a careful and thorough manner to investigate the CRISPR system in P. aeruginosa PAO1.

Nevertheless, we appreciate your kind suggestion and we have included in our research two additional strains of P. aeruginosa, PA14 and PA83, to enhance the robustness of our investigation. This information was incorporated on the subsection Validation of CRISPR-Cas system and DR length analysis (page 4, lines 95-103), on Results section (page 7, lines 146-154), and Discussion section (page 10, lines 220-228). A new dataset (namely S2 Dataset) was included with the data of these assemblies of P. aeruginosa strains obtained.

In addition to using the four most up-to-date bioinformatic tools mentioned in the previous section, the validation of the CRISPR-Cas system identification was conducted with two well-annotated CRISPR-Cas systems from P. aeruginosa strains PA14, which harbors a CRISPR-Cas type I-F, and P. aeruginosa PA83, which harbors both types I-F and IV-A, attributed to its chromosome and plasmid, respectively [15,21,22]. Following the procedure previously described, their genomes were obtained from NCBI [20] (https://www.ncbi.nlm.nih.gov/datasets/genome/; accessed on September 2, 2024) with GenBank accession numbers GCA_000014625.1 and CP017293.1, respectively.

4. While this study provides a computational analysis of CRISPR-Cas systems, it would be beneficial to consider future experimental validation (e.g., PCR) to confirm the predictions, which would enhance the credibility of the findings.

We appreciate and agree with your suggestion that further experimental validation would strengthen our findings. As you mentioned, this research focuses on a computational analysis that identifies an orphan CRISPR system in P. aeruginosa PAO1. As we highlighted in the manuscript (page 1, lines 26-28 and page 12, lines 302-306) we emphasize that our findings pave the way for future experimental research aimed at elucidating the functional role of the orphan CRISPR system in P. aeruginosa PAO1. We agree that such future work will be essential for validating and expanding upon the hypotheses made in this study.

---

## [Decision Letter · Decision Letter 1]

14 Oct 2024

Unveiling the Endogenous CRISPR-Cas System in * Pseudomonas aeruginosa*  PAO1.

PONE-D-24-28894R1

Dear Dr. Gonzalez-Reynoso,

We’re pleased to inform you that your manuscript has been judged scientifically suitable for publication and will be formally accepted for publication once it meets all outstanding technical requirements.

Kind regards,

Yanbin Yin

Academic Editor

PLOS ONE

Additional Editor Comments (optional):

Reviewers' comments:

Reviewer's Responses to Questions

**Comments to the Author**

1. If the authors have adequately addressed your comments raised in a previous round of review and you feel that this manuscript is now acceptable for publication, you may indicate that here to bypass the “Comments to the Author” section, enter your conflict of interest statement in the “Confidential to Editor” section, and submit your "Accept" recommendation.

Reviewer #1: All comments have been addressed

Reviewer #2: All comments have been addressed

2. Is the manuscript technically sound, and do the data support the conclusions?

Reviewer #1: Yes

Reviewer #2: (No Response)

3. Has the statistical analysis been performed appropriately and rigorously? 

Reviewer #1: Yes

Reviewer #2: (No Response)

4. Have the authors made all data underlying the findings in their manuscript fully available?

Reviewer #1: Yes

Reviewer #2: (No Response)

5. Is the manuscript presented in an intelligible fashion and written in standard English?

Reviewer #1: Yes

Reviewer #2: (No Response)

6. Review Comments to the Author

Reviewer #1: (No Response)

Reviewer #2: The authors have addressed all of the major concerns raised in the initial review, and the addition of validation part strengthens the paper's overall credibility.

7. PLOS authors have the option to publish the peer review history of their article (what does this mean?). If published, this will include your full peer review and any attached files.

Reviewer #1: No

Reviewer #2: No

---

## [Editor Report · Acceptance letter]

17 Oct 2024

PONE-D-24-28894R1 

PLOS ONE

Dear Dr. Gonzalez-Reynoso, 

I'm pleased to inform you that your manuscript has been deemed suitable for publication in PLOS ONE. Congratulations! Your manuscript is now being handed over to our production team.

Kind regards, 

on behalf of

Dr. Yanbin Yin 

Academic Editor

PLOS ONE
